# Work Stress and Willingness of Nursing Aides during the COVID-19 Pandemic

**DOI:** 10.3390/healthcare10081446

**Published:** 2022-08-02

**Authors:** Ting-Shan Chang, Li-Ju Chen, Shu-Wen Hung, Yi-Min Hsu, Ya-Ling Tzeng, Ying Chang

**Affiliations:** 1School of Nursing, Fooyin University, Kaohsiung 831301, Taiwan; aa332@fy.edu.tw; 2Department of Nursing, China Medical University Hospital, Taichung 404327, Taiwan; a10092@mail.cmuh.org.tw (L.-J.C.); n32638@mail.cmuh.org.tw (S.-W.H.); n4006@mail.cmuh.org.tw (Y.-M.H.); 3School of Nursing, College of Healthcare, China Medical University, Taichung 404328, Taiwan; tyaling@mail.cmu.edu.tw; 4Administration Department, China Medical University Hospital, Taichung 404327, Taiwan

**Keywords:** COVID-19, nurse aides, service quality, work stress

## Abstract

Objectives: During the coronavirus disease 2019 (COVID-19) pandemic, nursing aides (NAs) experienced greater work stress than they do typically because they worked in highly contagious environments. This may have influenced their work morale and willingness to work, which can reduce patient satisfaction, influence their physical and mental health, and even endanger patient safety or cause medical system collapse. Design: A cross-sectional survey with a structured self-report questionnaire was conducted. Setting and Participants: 144 NAs from a medical center in Central Taiwan participated. Methods: We recruited NAs through convenience sampling to discuss their work stress, willingness to work, and patients’ satisfaction with them during the COVID-19 pandemic. Result: Of the 144 recruited NAs, 115 (79.9%) were women and 29 (20.1%) were men, and 89 (61.8%) had completed COVID-19 training courses. NAs with different work tenure lengths exhibited significant differences in work stress (*p* = 0.022), willingness to work (*p* = 0.029), and patient satisfaction (*p* = 0.029) scores during the pandemic. Conclusion: The study findings provide crucial data for the management of NAs during pandemics to prevent them from neglecting patients due to excessive work stress or losing their willingness to work, which may cause the medical system to collapse.

## 1. Introduction

On 12 February 2020, the World Health Organization officially named the novel coronavirus coronavirus disease 2019 (COVID-19) [1,2,3]. Since its initial outbreak in Wuhan in late 2019, the COVID-19 pandemic has spread globally [4]. By May 2020, the pandemic had affected 184 countries, infected 3.2 million people, and killed approximately 0.23 million people, resulting in a fatality rate of 7.1%. In Taiwan, the Ministry of Health and Welfare had reported 351 confirmed cases and 6 fatalities from COVID-19. Of the confirmed cases, 27 were local, 314 were imported, and 10 were naval crew members of the *Panshi* fast combat support ship. This indicates the occurrence of sporadic community spread without larger-scale community spread or group infections in Taiwan, and that most confirmed cases were imported [5]. At the date of writing, the COVID-19 pandemic remains a global crisis. Many countries are still beset with the challenges posed by the pandemic. Given the highly contagious nature of the virus, community transmissions may take place rapidly if the necessary measures are not in place to contain the infection, resulting in a significant increase in the strain on healthcare capacity and death toll [6], and exposing healthcare workers to high risks of infection.

Recent changes in family structures in Taiwan, mainly caused by population aging and low birthrates, and the rapid evolution of diseases have resulted in a shortage of nursing human resources [7,8]. The discussion of nursing human resources and their supplementation is crucial. In consideration of organization re-engineering and cost efficiency, the role of assistants in nursing (hereafter nursing aides, NAs) was created to supplement nursing human resources. In Taiwanese medical institutions, NAs are second only to nursing personnel in terms of their interaction time and close relationships with patients. Relative to the prepandemic period, NAs during the current pandemic are required to work in highly infectious environments, which causes them to experience immense stress. Work stress influences one’s willingness to work. According to a study conducted after the outbreak of COVID-19, 61% of healthcare workers had fears of contracting the virus through contact with patients, 43% reported work overload, and 49% experienced stress due to job burnout [9]. Many studies have identified the shortage of caregiving staff as the main stressor during the pandemic [10]. The shortage of staff has put NAs under persistently high levels of stress and, in turn, dampened their willingness to work, leading to an ever-growing staff turnover [11].

An overburden of stress causes fatigue [12,13], which reduces one’s willingness to work and work efficiency. High stress among medical personnel reduces patient satisfaction [14], in turn causing NAs to neglect their duties, become emotionally unstable, experience career burnout, and ultimately resign [7,15]. Therefore, pandemic-related stress among NAs is worthy of concern. During the pandemic, NAs were worried and fear that they or their families will become infected, driving them to refuse to provide clinical care for patients. Under severe pandemic conditions, the number of cases will increase, causing the quantity of nursing personnel and human resources to be insufficient. This increases the demand for NAs in clinical care institutions to account for the increased number of patients. In these circumstances, if all NAs refuse to provide clinical care to patients, the entire healthcare system will inevitably collapse, thereby affecting various aspects of society. Throughout the pandemic, few scholars have focused on the work stress of NAs. However, the role of NAs in medical institutions should not be overlooked; understanding the work stress and willingness to work of NAs during the pandemic is urgently required. This study examined the work stress, willingness to work, and patient satisfaction regarding NAs during the COVID-19 pandemic.

## 2. Literature Review

NAs are members of healthcare and nursing teams whose role and function is to perform basic and simple nursing responsibilities under the supervision of nursing professionals; these responsibilities generally involve body cleaning, comfort nursing, excretory care, dietary assistance, supervised tube feeding, and patients’ comfort and mobility [16]. In addition to replacing the role of accompanying family members, NAs also serve other functions such as assisting nursing staff, performing daily care tasks for patients, bridging the tripartite communication of information among patients, family members, and healthcare workers, and providing mental support [17]. Work stress refers to a situation and process in which the negative feelings arising from maladaptation to work cause a worker to change their physiological or psychological conditions in such a way that they deviate from normal functioning [18]. Owing to the high-stress nature of their work, NAs are persistently exposed to high stress levels, which may lead to problems such as neglect of duty, emotional instability, job burnout, and turnover. During the pandemic, due to a lack of understanding about the new disease, its highly contagious nature, and the fact that most NAs were unable to work when placed in quarantine, the NAs experienced work overload, fear of contagion for themselves and their loved ones, and substantial work stress [19]. Fear of stigmatization also causes workers in the healthcare field to be hesitant about caring for patients on the frontline [18,20]. Any emergency that involves infections, such as the COVID-19 pandemic, may alter healthcare workers’ willingness to work for various reasons. The potential ethical dilemma they face lies mainly in how to balance one’s ethical duty to care for patients with concerns of contracting the COVID-19 virus and spreading it to their patients and family members. Similar concerns may be compounded by the limited availability of personal protective equipment, inequitable distribution of available equipment, and limited and constantly changing recommendations [21,22]. The above literature casts light on the role of NAs as workers who spent the longest time and had the closest relationship with patients in the healthcare environment during the pandemic. Their work stress and willingness to work are subjects worth prioritizing and exploring in great depth.

## 3. Materials and Methods

### 3.1. Study Design

A cross-sectional design was implemented, using self-report structured questionnaires for data collection and convenience sampling.

#### 3.1.1. Setting and Participants

The study was conducted from August 2020 to December 2020 at an urban medical center in central Taiwan. The following selection criteria were used: (a) worked as NAs in the medical center; (b) had spent more than 12 months in the same position and unit; and (c) possessed Chinese communication abilities. Of the 200 eligible participants, 144 NAs agreed to take part in the study and completed the questionnaires (response rate = 72%). The necessary sample size was estimated using the G-Power program prior to data collection [23]. The result of the analysis indicates a minimum number of 130 participants, considering the following parameters: an average effect size of 0.5, an alpha of 0.05, and power of 0.8.

#### 3.1.2. Ethical Considerations

This study was approved by the Institutional Review Board of China Medical University Hospital, Taiwan (CMUH109-REC1-076). Written informed consent was obtained from all NAs.

#### 3.1.3. Data Collection

To understand whether NAs’ quality of care is influenced by work stress, willingness to work, and patient satisfaction, this study also collected questionnaires on patient satisfaction, routinely obtained by NAs in the past, mainly from the units and hospitalized patients under their care. To ensure the accuracy of the data, the questionnaires were collected anonymously. We recruited nurse aides who met the inclusion criteria and provided them with all necessary information, such as the purpose of the study, the procedures involved, and the conditions of anonymity, confidentiality, and voluntary participation. A hard copy of the questionnaire was distributed after NAs signed the participant consent form. In order to understand the impact of the pandemic on NAs, we asked the NAs to fill out the questionnaires on work stress and willingness to work before and after the pandemic. When distributing the questionnaires, the instructions were clearly stated, and the questionnaires were clearly marked to distinguish between before or after the pandemic. The questionnaire required approximately 20 min to complete. Of the 150 questionnaires distributed in the present study, 144 were retrieved, yielding a response rate of 96%.

#### 3.1.4. Measures

The structured questionnaires consisted of scales for work stress, willingness to work, and patient satisfaction. The questionnaire was developed through rigorous steps to ensure its reliability and validity. In particular, the work stress and willingness to work scales were self-reported by the NAs, and the patient satisfaction scale was completed by patients or their families. The scales and their corresponding content in the questionnaire are briefly detailed as follows:

#### 3.1.5. Demographic Characteristics

The questionnaire collected information on age, sex, marital status, education level, work tenure, salary, religion, and participation in NA courses or COVID-19 courses.

#### 3.1.6. Work Stress Scale

The questionnaire comprised 26 questions on work execution, workload, complaints about mistakes, income and benefits, performance evaluation system, and career advancement training. For example: 1. execution of nursing tasks, 2. variations in patient conditions, 3. needs of patients, 4. needs of family members, 5. records of patient conditions, and 6. work shifts. Responses were captured on a 5-point Likert scale (1 = “very light stress”; 5 = “very heavy stress”). Higher scores were considered to indicate higher stress levels. An example item is outlined as follows: “Performing caregiving work and monitoring disease changes in patients.” In this study, the Cronbach’s α value for this scale was 0.849–0.934 [24].

#### 3.1.7. Willingness to Work Scale

This questionnaire included five items on willingness to work to evaluate participants’ future willingness to work on the basis of a single behavioral aspect (Liu, 2017). For example, 1. Are you continuing your care work during the COVID-19 pandemic? 2. Will you continue your care work after the COVID-19 pandemic? 3. I will strive to overcome various difficulties and work for at least 1 year during the pandemic. 4. I will strive to overcome various difficulties and continue to work as a nursing assistant for 1 year. 5. I will strive to overcome various difficulties and intend to stay in the hospital to work as a nursing assistant. Responses were obtained on a 5-point Likert scale (1 = “very unwilling”; 2 = “unwilling”; 3 = “neutral”; 4 = “willing”; and 5 = “very willing”) Higher scores were considered to indicate higher willingness to work. An example item is outlined as follows: “Will you work as a nursing assistant during the COVID-19 pandemic?” In this study, the Cronbach’s α value for this scale was 0.924.

#### 3.1.8. Patients’ Satisfaction Scale

The questionnaire comprised a self-developed 10-item scale divided into two sections. Section one comprised five items on overall care provided by the NA. An example item of this questionnaire is outlined as follows: “The NA noticed the patients’ needs and actively provided assistance.” Section two comprised five items on the behaviors and actions of the NA. An example item of this questionnaire is outlined as follows: “The NA has a friendly attitude and is patient.” Responses were obtained on a 5-point Likert scale (1 = “very unsatisfied”; 2 = “unsatisfied”; 3 = “adequate”; 4 = “satisfied”; and 5 = “very satisfied”). Total scores ranged from 10 to 50 points, with a higher score implying greater service satisfaction. Patients who were receiving care or their family members responded to the questionnaire. The Cronbach’s α for the scale was 0.886.

### 3.2. Data Analysis

All statistical analyses were performed using SPSS (version 25.0, SPSS Inc. Chicago, IL, USA). We initially obtained descriptive statistics to identify patient characteristics. An independent sample t test and one-way analysis of variance were performed to determine differences in work stress, willingness to work, and patient satisfaction among the NAs. Subsequently, we adopted a paired t test to identify differences in these three variables between before and during the pandemic. Finally, Pearson’s correlation coefficient analysis was employed to identify correlations among the variables. All reported *p* values are two-tailed, and statistical significance was set to 0.05.

## 4. Results

The research participants consisted of 29 (20.1%) men and 115 (79.9%) women. Most of the participants (56.3%) were aged 51–60 years, 80 (55.6%) were married, and 68 (47.2%) had a senior or vocational high school education level. Most of the participants had 6–10 years of work experience in the care industry (63; 43.8%), earned a salary of TWD 30,001–40,000 per month (59; 41.0%), were Buddhist (79; 54.9%), and participated in COVID-19 courses (89; 61.8%; Table 1).

Further analysis with Scheffé’s method revealed that NAs with 20 years or more of work tenure experienced the highest work stress, and NAs with different work tenures experienced significantly different levels of work stress during the pandemic (*p* = 0.022; Table 2). Additionally, the participants’ work stress scores (52.37 ± 19.98) before the pandemic were significantly lower than during the pandemic (68.77 ± 24.76; *p* = 0.000; Table 3). NAs with longer work tenures are relatively older and possibly have limited physical energy; however, as no data were collected on their health or physical energy, the correlation involved requires further discussion.

In addition to providing clinical care for patients during the pandemic, NAs with long work tenures had to follow pandemic control measures established by the government and their hospitals. Because their physical energy and memory were unable to meet expectations, NAs with long work tenures experienced high work stress during the pandemic.

Table 2 indicates that NAs with 11–15 years of work tenure were the most willing to work, and NAs with different work tenures had significantly different degrees of willingness to work during the pandemic (*p* = 0.029; Table 2). At this career stage, NAs have accumulated sufficient work experience and are in a relatively fit physical condition. Therefore, they were confidently able to adhere to pandemic control measures established by the Taiwan government and hospitals when providing clinical care to patients. Additionally, the willingness to work scores were significantly higher prior to the pandemic (19.72 ± 3.44) compared with during the pandemic (18.72 ± 3.87; *p* = 0.000; Table 3). Although NAs were not assigned to provide clinical care for patients with confirmed or suspected COVID-19, when providing care for other patients, NAs were required to adhere to pandemic control measures, and they worried whether the caregiving process can transmit COVID-19. Therefore, NAs were less willing to work during the pandemic than they were before the pandemic.

The patient satisfaction scores were slightly higher during the pandemic (46.56 ± 4.59) than they were before the pandemic (46.27 ± 5.53). However, this difference was nonsignificant (*p* = 0.633; Table 3). Because patients and families observed that NAs endeavored to provide the same amount and quality of care as that provided in nonpandemic periods, the patients’ satisfaction with NA before and during the pandemic did not differ. Pearson’s correlation coefficient analysis indicated a significant negative correlation between work stress and willingness to work; high work stress resulted in a lower willingness to work (*p* = 0.042; Table 4).

## 5. Discussion

This study has considerable research value; it investigated the work stress and willingness to work of Taiwanese NAs and patient satisfaction with them during the COVID-19 pandemic. Relative to before the pandemic, NAs experienced higher work stress during the pandemic. This result is consistent with that of a 2020 study on the work stress experienced by 314 nursing personnel during the COVID-19 outbreak [25]. It was a qualitative study that was conducted using semi-structured interviews with 97 healthcare professionals [26]. The findings are also consistent with a 2021 study carried out by Elena Fiabane on 728 healthcare workers.

Our findings reveal that NAs were less willing to work during the COVID-19 pandemic compared with nonpandemic periods. This result is consistent with that of a previous study on the rapid spread of SARS that demonstrated that nursing personnel experienced immense stress and refused to provide care for patients, which reduced their willingness to work and motivated them to resign [15]. Our results are also consistent with the findings of a study on post-traumatic stress and resignation intentions among 300 nursing personnel in the Middle East during the Middle East respiratory syndrome outbreak. During outbreaks of infectious diseases, medical personnel have a lower willingness to work and increased resignation intentions [27]. Moreover, our results do not reveal significant differences in patient satisfaction with NAs before and during the COVID-19 pandemic. This implies that although NAs experienced greater work stress and were less willing to work during the pandemic, the pandemic did not influence patient satisfaction with NAs. The findings highlight that Taiwan’s experience and lessons learned from managing the SARS outbreak facilitated the establishment of effective NA management and training policies to provide patient care during the outbreak of infectious diseases. This motivated NAs in Taiwan to continue providing satisfactory-quality services during the COVID-19 pandemic.

Our findings reveal significant differences in the work stress and willingness to work among NAs with different lengths of work tenure; specifically, NAs with longer work tenures experienced higher work stress and lower willingness to work. This suggests that during outbreaks of infectious diseases, NAs with more experience endure more work stress and are less willing to work at healthcare institutions. Possible reasons for this include the older age of NAs with 20 years of work tenure. By contrast, NAs with 11–15 years of work tenure have sufficient work experience but more physical energy than older NAs do. A literature review revealed that most studies have not included work tenure in their analyses. Future studies should include qualitative interviews to further clarify relevant factors that cause work tenure to influence work stress and willingness to work during a pandemic.

Despite Taiwan’s comprehensive pandemic control policies, by December 2020, the pandemic had affected 191 countries, infected 76 million people, and killed approximately 1.69 million people, resulting in a fatality rate of 2.23%. The COVID-19 pandemic has subjected both frontline care providers and medical staff to high levels of stress. This study emphasizes the necessity of strengthening the infectious disease education and training of NAs. Although basic training courses for infectious disease control have been introduced in NA training courses, the training provided in these courses is insufficient for pandemic management. Therefore, future in-service training courses should include a required number of training hours on infectious disease education. In response to the limited knowledge on COVID-19 and its highly contagious nature, a COVID-19 pandemic training program that emphasizes the coping abilities of the caregiving team and provides suitable training content should be established to increase the effectiveness of care for patients with COVID-19 and protect the caregivers. By using multimedia and internet platforms, healthcare institutions can promote the participation and training of all staff members, thereby enhancing caregivers’ knowledge and skill capacities as well as their abilities to cope with the pandemic [28,29,30].

The COVID-19 pandemic has had a worldwide impact, but the effects on healthcare systems have varied with the different stages of the pandemic and disparities in pandemic containment policies. Gathering the experiences from different regions during the different stages of the pandemic can help increase the understanding of the global pandemic and, in turn, facilitate the discussion and formulation of more effective coping solutions. According to the results of this study, although Taiwan had a well-developed set of pandemic containment policies in place and managed to keep the situation under control during the early stage of the outbreak, the pandemic still created immense pressure on NAs tending to patients on the frontline. Hence, it is imperative to offer suitable support measures for frontline workers to reduce the impact of the pandemic on their work [30]. Other studies also pointed out that proactive coping measures adopted by the government and hospital management during the pandemic can alleviate the stress experienced by employees and recommended that suitable measures be adopted to improve healthcare personnel’s willingness to work [31,32].

The results of this study have underscored the imperative need to strengthen education and training on infectious diseases for NAs and shed light on the fact that although training in infectious disease control and prevention has been incorporated into the basic curriculum of vocational education programs for NAs, it remains inadequate in the face of the pandemic. Thus, future in-service education and training programs should include more learning hours on infectious diseases. Owing to the limited knowledge of COVID-19 and its highly contagious nature, to provide better care for patients and protect the nursing staff, it is important to strengthen training of the nursing team’s coping capabilities, formulate training plans for the pandemic, and set up training content rationally. Multimedia online platforms should also be leveraged to promote the participation of all staff in training and improve nursing staff’s knowledge, skills reserve, and pandemic response capabilities [18]. In addition, managers should allow NAs to relieve their stress, whenever appropriate, by actively providing them with mental support, spiritual support group sessions, and pandemic protection measures. Retention or reward mechanisms should also be put in place to improve the lack of willingness to work in certain staff. Moreover, a sufficient level of nursing staff should be maintained in clinical practice to avoid the problem of staff turnover caused by work overload.

## 6. Limitations

Because this study investigated only one medical center in Central Taiwan, the results cannot be generalized to other regions in Taiwan. Future studies should increase the sample size and sample NAs from various regions to increase the generalizability of the results. However, the investigated medical center has 2185 beds and serves more than 2.88 million patients annually. Although the results cannot be applied to other parts of Taiwan, the demographic characteristics of the recruited participants are in line with the national NA statistics of 2057 males and 11,700 females; they are thus representative to an extent. As a quantitative study, this study did not explore how the pandemic influences the work stress and willingness to work of NAs or investigate factors related to changes in patient satisfaction. Future studies should include qualitative interviews for greater comprehensiveness.

## 7. Conclusions

This study provided critical information on the NAs’ work stress, willingness to work, and patient satisfaction with them during the COVID-19 pandemic. The results reveal that during a pandemic, NAs experience high work stress and low willingness to work. However, patient satisfaction was not affected. This was because NAs earn recognition from patients and family members by providing consistent professional care before and during the pandemic. Accordingly, institutions should increase the depth of courses on infectious disease control to establish a psychological support system for medical personnel and prepare them for disease outbreaks. This can mitigate the immense stress experienced by frontline medical personnel during pandemics; this support can positively influence their physical and mental status, in turn preventing reductions in their willingness to work, and ultimately averting the collapse of healthcare systems during a pandemic.

## Figures and Tables

**Table 1 healthcare-10-01446-t001:** Demographic distribution of the participants (n = 144).

Variable		Frequency	Percentage (%)
Sex			
	Male	29	20.1
	Female	115	79.9
Age (years)			
	≤40	12	8.3
	41–50	29	20.1
	51–60	81	56.3
	≥61	22	15.3
Marital status			
	Unmarried	25	17.4
	Married	80	55.6
	Divorced	26	18.1
	Widowed	13	9.0
Education level			
	Undereducated/Elementary school	24	16.7
	Junior high school	29	20.1
	Senior or vocational high school	68	47.2
	Junior college	20	13.9
	College	3	2.1
NA work tenure			
	≤5 years	39	27.1
	6–10 years	63	43.8
	11–15 years	24	16.7
	16–20 years	14	9.7
	≥20 years	4	2.8
Salary			
	TWD 10,000–20,000	10	6.9
	TWD 20,001–30,000	52	36.1
	TWD 30,001–40,000	59	41.0
	TWD 40,001–50,000	15	10.4
	≥ TWD 50,001	8	5.6
Religion			
	Buddhism	79	54.9
	Christianity	11	7.6
	Others	10	6.9
	None	44	30.6
Participation in NA courses			
	Yes	123	85.4
	No	21	14.6
Participation in COVID-19 courses			
	Yes	89	61.8
	No	55	38.2

**Table 2 healthcare-10-01446-t002:** Differences in work stress, willingness to work, and patient satisfaction with NAs during the COVID-19 pandemic by demographic variables.

Variable	Work Stress			Willingness to Work			Patient Satisfaction		
	Mean (SD)	t/F	*p*	Mean (SD)	t/F	*p*	Mean (SD)	t/F	*p*
Sex		0.198	0.843		−0.318	0.751		0.446	0.656
Men	69.59 (21.9)			18.52 (3.7)			46.90 (4.9)		
Women	68.57 (25.5)			18.77 (3.9)			46.47 (4.5)		
Age (years)		0.624	0.601		1.094	0.354		0.954	0.416
≤40	77.75 (29.26)			17.58 (4.1)			46.33 (4.3)		
41–50	69.24 (25.2)			18.31 (3.8)			47.79 (3.5)		
51–60	67.91 (25.0)			18.73 (3.7)			46.35 (4.9)		
≥61	66.41 (21.27)			19.86 (4.5)			45.82 (4.8)		
Marital status		0.729	0.536		2.012	0.115		0.606	0.612
Unmarried	74.28 (26.3)			18.56 (4.7)			46.92 (4.8)		
Married	71.04 (23.5)			18.20 (3.4)			46.19 (4.7)		
Divorced	62.23 (25.8)			19.50 (3.8)			46.65 (4.6)		
Widowed	57.31 (23.7)			20.69 (4.6)			47.92 (3.8)		
Education level		0.197	0.939		0.639	0.636		0.497	0.738
Elementary school	72.46 (21.2)			18.67 (4.7)			46.21 (4.9)		
Junior high school	66.86 (21.6)			17.86 (3.8)			46.59 (4.2)		
Senior or vocational high school	68.81 (27.6)			19.01 (3.5)			46.62 (4.7)		
Junior college	67.00 (23.1)			19.25 (4.0)			47.20 (4.8)		
College	68.67 (36.1)			17.33 (4.6)			43.33 (3.5)		
Nursing aide work tenure		2.954	0.022		2.788	0.029		0.573	0.682
≤5 years	71.23 (27.5)			19.26 (3.8)			46.95 (4.4)		
6–10 years	70.19 (21.5)			17.92 (3.7)			46.10 (4.7)		
11–15 years	59.13 (25.3)			20.38 (3.7) *			47.42 (4.4)		
16–20 years	63.21 (25.0)			18.93 (3.8)			45.79 (5.2)		
>20 years	99.75 (14.8) *			15.50 (5.3)			46.56 (4.6)		
Salary		0.786	0.536		0.741	0.566		1.154	0.334
TWD 10,000–20,000	67.4 (22.6)			18.60 (3.2)			48.10 (3.2)		
TWD 20,001–30,000	68.44 (26.9)			18.37 (3.7)			46.62 (4.6)		
TWD 30,001–40,000	71.27 (21.7)			18.61 (4.0)			46.64 (4.7)		
TWD 40,001–50,000	68.40 (30.7)			19.47 (4.3)			44.47 (4.8)		
≥TWD 50,001	54.88 (23.1)			20.63 (4.1)			47.50 (4.1)		
Religion		0.175	0.913		0.722	0.541		0.851	0.468
Buddhism	69.37 (25.9)			18.68 (4.0)			46.39 (5.0)		
Christianity	63.91 (20.6)			17.27 (2.5)			46.36 (4.0)		
Others	67.10 (20.5)			18.60 (3.6)			48.80 (3.2)		
None	69.30 (25.0)			19.18 (4.0)			46.39 (4.3)		
Participation in NA courses		1.260	0.210		0.010	0.992		−0.222	0.825
Yes	67.70 (24.9)			18.72 (3.8)			46.52 (4.6)		
No	75.05 (23.6)			18.71 (4.4)			46.76 (4.4)		
Participation in COVID-19 courses		0.931	0.354		−1.345	0.181		1.292	0.18
Yes	70.28 (24.13)			18.38 (4.0)			46.94 (4.2)		
No	66.33 (25.8)			19.27 (3.6)			45.93 (5.1)		

* *p* < 0.05

**Table 3 healthcare-10-01446-t003:** Work stress, willingness to work, and patient satisfaction with NAs before and during the pandemic (n = 144).

Variable	Before the Pandemic	During the Pandemic	t
	Mean (SD)	Mean (SD)	
Work stress	52.37 (19.98)	68.77 (24.76)	−10.04 (0.000)
Willingness to work	19.72 (3.44)	18.72 (3.87)	4.50 (0.000)
Patient satisfaction	46.27 (5.53)	46.56 (4.59)	−0.478 (0.633)
Paired *t* test			

**Table 4 healthcare-10-01446-t004:** Correlation analysis among work stress, willingness to work, and patient satisfaction with NAs (n = 144).

Variable	Work Stress	Willingness to Work	Patients’ Satisfaction
Work stress	1	0.042 *	0.280
Willingness to work	0.042 *	1	0.579
Patient satisfaction	0.280	0.579	1

* *p* < 0.05.

## Data Availability

Not applicable.

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
