# Peer review of "Work Stress and Willingness of Nursing Aides during the COVID-19 Pandemic"

_healthcare, 2022, doi:10.3390/healthcare10081446_

Round 1

Reviewer 1 Report

I want to congratulate the authors on the subject of the manuscript. In my opinion, I found the study very interesting, and I think the topic is very necessary. The manuscript is written in an understandable way and contains in each section the most relevant aspects of the research. However, the manuscript must make some important revisions:

-Suitable title, abstract and keywords. Approx. 15 words.

-Introduction: The introduction should be improved to provide sufficient international background. The research problem is important and current. Update references 10,11,12.

-Material and methods: The design and type of study would need to be better defined, the type of sampling, types of variables, the steps of the data collection procedure, duration of the study. The characteristics and sample size of the population: rural or urban. Validation, adaptation, translation and reliability of each instrument (or dimensions of the questionnaire) used.

-Results: Correct presentation and design of the results. The choice of statistical processes for the analysis is adequate. Only sometimes data is repeated in the text and in the table. There is talk of a cross-sectional survey, but table 3 shows the results before and during the pandemic (¿quasi-experimental...?).

-Discussion: The order in the presentation of the discussion is appropriate with the results. There are some claims and lack of robustness or strengths in the discussion both in their data and in international references.

-References: Updated, although few citations and international vision.

Author Response

Dear Reviewer:
Please refer to the attachment for the suggested reply, thank you

Reviewer 2 Report

First of all, the topic is good, however, there are so many similar works. You need to revise the conclusion section by stating the key contribution of this study. Also, the number of references are very few. You used only 17 references. You also have no literature review section. Please add a separate part on literature review. 

In the appendix section, please add the questionnaire. Mention the managerial and practical significance of the study. Good luck. 

Author Response

(The authors gave the same response as above.)

Reviewer 3 Report

Thank you for the chance to review this paper which explores the work stress and willingness of Nursing Aides to work during COVID 19 pandemic.

There are some points in the paper which need addressing

The method do not state how patients and families were recruited yet their results are presented

The results refer to data pre pandemic, but it is not clear when or how this data was collected.

Only one measurement tool was a validated measure so what actions were taken to explore validity and reliability of tools used?

Is it too much of an assumption to claim that NAs with longer tenure had reduced physical energy as no data was collected on their health or physical energy?

Author Response

(The authors gave the same response as above.)

Round 2

Reviewer 1 Report

I want to congratulate the authors for the theme of the manuscript. In my opinion, I found the study very interesting and I think the topic is very necessary. After review, the manuscript is written in an understandable way and contains the most relevant aspects of the research in each section.

Author Response

DEAR Reviewer:
Thank you very much for your affirmation and encouragement

Reviewer 2 Report

The revised version is ok. 

Author Response

(The authors gave the same response as above.)

Reviewer 3 Report

In the willingness to work section the questions do not seem coherent. I understood from recruitment the NAs were in employment yet the willingness to work question example is “Will you work as a nursing assistant during the COVID-19 pandemic?”  The questionnaire has been added as a supplementary file but is not accessible to international readers. May just a text box with the questions would suffice

The addition this statement still needs clarifying ‘we asked the  NAs to fill out the questionnaires on work stress and willingness to work before and after the pandemicas data was collected during the pandemic august 2020 to December 2020. Were you asking participants to reflect on their willingness to work pre pandemic if so this needs to be clear and discussed as a limitations due to recall bias. Again site of all questions will help the reader.

I think it is a use of language  for example this sentence suggest data collected before the pandemic ‘Additionally, the willing ness to work scores were significantly higher prior to the pandemic (19.72 ± 3.44) compared with during the pandemic (18.72 ± 3.87; P = .000; Table 3)But is the meaning NAs reported higher willingness to work scores when ask about this in relation to their work before the pandemic.

Spelling mistake in abstract line 20 should read completed

The supplementary file added on the questionnaire is not accessible to an international reader audience

Author Response

DEAR Reviewer:
Thank you for your suggestion, reply as attachment

This manuscript is a resubmission of an earlier submission. The following is a list of the peer review reports and author responses from that submission.